

**Mixing-layer-height-referenced ozone vertical distribution in the lower troposphere of Chinese megacities: Stratification, classification, meteorological, and photochemical mechanisms**

Zhiheng Liao[a, b], Meng Gao[c], Jinqiang Zhang[d, e], Jiaren Sun[f], Jiannong Quan[a], Xingcan Jia[a], Yubing Pan[a], Shaojia Fan[b, g]

[a] Institute of Urban Meteorology, Chinese Meteorological Administration, Beijing, China

[b] School of Atmospheric Sciences, Sun Yat-Sen University, Zhuhai, China

[c] Department of Geography, Hong Kong Baptist University, Hong Kong SAR, China

[d] Key Laboratory of Middle Atmosphere and Global Environment Observation, Institute of Atmospheric Physics,
Chinese Academy of Sciences, Beijing, China

[e] College of Earth and Planetary Sciences, University of Chinese Academy of Sciences, Beijing 100049, China

[f] Key Laboratory of Urban Ecological Environmental Simulation and Protection of Ministry of Environmental
Protection, South China Institute of Environmental Sciences, Ministry of Ecology and Environment of the PRC,
Guangzhou, China

[g] Guangdong Provincial Observation and Research Station for Climate Environment and Air Quality Change in the
Pearl River Estuary, Key Laboratory of Tropical Atmosphere–Ocean System, Ministry of Education, Southern
Marine Science and Engineering Guangdong Laboratory (Zhuhai), Zhuhai, China

**Corresponding author**: S. J. Fan (eesfsj@mail.sysu.edu.cn)

**Abstract:** Traditional tropospheric ozone ($O_3$) climatology uses a simple average substantially smoothed stratification structure in individual $O_3$ profiles, limiting our ability to properly describe and understand how $O_3$ is vertically distributed at the interface between the mixing layer (ML) and free troposphere (FT). In this study, we collected 1,897 ozonesonde profiles from two Chinese megacities (Beijing and Hong Kong) over the period 2000–2022 to investigate climatological vertical heterogeneity of lower-tropospheric $O_3$ distribution with a
mixing-layer-height-referenced ($h$-referenced) vertical coordinate system. The mixing layer height ($h$) was first estimated following an integral method that integrates the information of temperature, humidity, and cloud. After that, a so-called $h$-referenced vertical distribution of $O_3$ was determined by averaging all individual profiles expressed as a function of z/h rather than z (where z is altitude). We found that the vertical stratification of $O_3$ is distributed heterogeneously in the lower troposphere, with stronger vertical gradients at the surface layer and ML–
FT interface. There are low vertical autocorrelations of $O_3$ between the ML and FT, but high autocorrelations within each of the two atmospheric compartments. These results suggest that the ML–FT interface acts as a geophysical "barrier" separating air masses of distinct $O_3$ loadings. This barrier effect varies with season and city, with an ML-to-FT detrainment barrier in summer (autumn) and an FT-to-ML entrainment barrier in other seasons in Beijing (Hong Kong). Based on Student's $t$ test, $h$-referenced $O_3$ profiles were further classified into three typical patterns:
$MLO_3$-dominated, $FTO_3$-dominated, and uniform distribution. Although the $FTO_3$-dominated pattern occurs most frequently during the whole study period (69% and 54% of days in Beijing and Hong Kong, respectively), the $MLO_3$-dominated pattern prevails in the photochemical active season, accounting for 47% of summer days in Beijing and 54% of autumn days in Hong Kong. These occurrences of the $MLO_3$-dominated pattern are significantly more frequent than in previously reported results at northern mid-latitudes, indicating intensive
photochemical $MLO_3$ production under the high-emission background of Chinese megacity. From $FTO_3$-dominated to $MLO_3$-dominated pattern, the $O_3$ precursor $CH_2O$ ($NO_2$) experiences a substantial increase (decrease) in Beijing,




but a slight increase (decrease) in Hong Kong. Vertically, the increment of $CH_2O$ is larger in the upper ML and the decrement of $NO_2$ is larger in the lower ML. Such changes in $O_3$ precursors push $O_3$ production sensitivity away from the VOC-limited regime and facilitate high-efficiency production of $O_3$ via photochemical reactions, particularly in the upper ML.

## 1 Introduction

Ozone ($O_3$), the dominant precursor of hydroxyl radicals, plays a crucial role in tropospheric chemistry. It is also an important greenhouse gas closely related to climate change and environmental issues (Seinfeld and Pandis, 2016;Monks et al., 2015). Being an air pollutant, $O_3$ can influence air quality on a hemispheric scale, exerting detrimental effects on human health and vegetation (Fleming et al., 2018;Mills et al., 2018). Tropospheric $O_3$ is primarily formed through a complex series of photochemical reactions between nitrogen oxides (NOx) and volatile organic compounds (VOCs) in the presence of sunlight (Seinfeld and Pandis, 2016). There are substantial emissions of NOx and VOCs in urban regions, where most of the population and industry are concentrated. As a result, elevated $O_3$ concentrations in the lower troposphere remain a persistent environmental problem in urban regions around the world (Lu et al., 2018). Significant efforts have been made to understand $O_3$ pollution in different cities (Monks et al., 2015). However, most previous studies were based on ground-based observations, and gave only limited insight into $O_3$ vertical distribution.

$O_3$ vertical distribution in the lower troposphere can provide very important information for mechanistic understanding of surface $O_3$ pollution (He et al., 2021;Lin et al., 2010;Jaffe, 2011;Yates et al., 2017). One of the major advantages when dealing with $O_3$ profile data is able to discriminate the two specific $O_3$ components corresponding to the two "reservoirs"—the mixing layer (ML) and the free troposphere (FT)—and therefore, to determine the direction and intension of vertical exchange processes across the ML-FT interface. Several studies have been made trying to resolve the $O_3$ vertical exchange problem in the lower troposphere (Neuman et al., 2012;Berkes et al., 2016;Kaser et al., 2017;Trousdell et al., 2016;Zhao et al., 2019;Lin et al., 2010;Zhu et al., 2020). For example, based on tethered ozone soundings during a four-day ozone episode in southern Taiwan, Lin et al. (2010) revealed that the increase rate of surface $O_3$ concentration due to the downward mixing of the $O_3$ from the $O_3$ reservoir layers can be as high as 12–24 ppbv h$^{-1}$ in the late morning. Based on 214 aircraft vertical profiles in Colorado during summer 2014, Kaser et al. (2017) investigated the $O_3$ vertical gradient between the ML and the FT in order to estimate the FT-to-ML $O_3$ entrainment and to evaluate its representation in the WRF-Chem model. Their study focusing on the $O_3$ entrainment highlighted deficiencies in the model, indicating an overestimation of the $O_3$ entrainment and a too-efficient vertical mixing in the lower ML. These deficiencies were found to originate mainly from errors in the entrainment rate and ML height during the morning and an erroneous representation of the $O_3$ gradient at the ML–FT interface during the rest of the day. Overall, by measuring the specific terms in the vertical $O_3$ budget, detailed comparisons with photochemical models can uncover distinct weaknesses in current models and discern whether the difficulties lie in dynamical (transport) or chemical aspects of the numerical efforts (Trousdell et al., 2016).

$O_3$ vertical stratification below and above the ML-FT interface (i.e., the mixing layer height, $h$) is the basis for ozone vertical exchange processes. The formation of $O_3$ stratification is mainly due to the fact that the turbulent-convective ML and overlaying FT are usually separated by the mixing layer capping inversion, which acts as a transport barrier (Donnell et al., 2001). This barrier is indicated by steep vertical gradients of meteorological variables and chemical constituents (Petetin et al., 2018;Wyngaard and Brost, 1984;Williams et al., 2011). This means that climatological $h$-referenced $O_3$ vertical distribution in the lower troposphere could provide a useful



reference for understanding vertical exchange processes and validating air quality numerical models. However, tropospheric $O_3$ climatology is traditionally formed in a sea-level-referenced vertical coordinate system (Ding et al., 2008;Liao et al., 2021;Diab et al., 2004;Yonemura et al., 2002;Stauffer et al., 2016). Owing to day-to-day variation in the mixing layer top height, vertical stratification introduced in all individual profiles can be substantially
smoothed in climatological profile when adopting the traditional vertical coordinate system. To address this issue, Petetin et al. (2018) proposed $h$-referenced climatology of lower-tropospheric $O_3$ profiles based on aircraft and ozonesondes at northern mid-latitudes over 1994–2016. When adopting this $h$-referenced vertical coordinate system, $O_3$ vertical stratification can be well preserved in lower-tropospheric $O_3$ climatology, demonstrating a significant improvement in capturing possible specific features (i.e., stratification) in the $O_3$ vertical distribution that would be
smoothed with a simple average, in particular at the ML–FT interface. However, the $h$-referenced $O_3$ climatology in Petetin et al. (2018) is a hemispheric-scale composite result, which cannot represent the state over polluted urban regions, including megacities.

O$_3$ pollution has long been a significant environmental issue in China, despite the 2013 Clean Air Action Plan. In
recent photochemical active seasons, $O_3$ overtook fine particles as the most important air pollutant in the three major city agglomerations: the North China Plain (NCP), the Yangtze River Delta (YRD), and the Pearl River Delta (PRD). As such, urban $O_3$ pollution is becoming a priority for scientific research and control strategies in China (Lu et al., 2018;Wang et al., 2022b), and numerous studies have explored the spatiotemporal characteristics and formation mechanisms of surface $O_3$ pollution, as summarized in Wang et al. (2017) and Wang et al. (2022b).
Moreover, there are ongoing efforts to understand the role of vertical exchange in surface $O_3$ pollution in China based on vertical observations from tower-based, tethered-balloon-based, unmanned-aerial-vehicle-based, aircraft-based, and lidar-based observations (Lin et al., 2010;Zhao et al., 2019;He et al., 2021;Benish et al., 2020;Zhu et al., 2020;Han et al., 2020;Chen et al., 2023). These vertically observational studies generally indicate that merging of the stable boundary layer, residual layer, and convection-driven mixing layer involves the mixing of
trace gases from these different atmospheric layers, and leads to complex vertical $O_3$ profiles. However, these existing $O_3$ vertical observations suffer from low observation height (tower-based, tethered-balloon-based, unmanned-aerial-vehicle-based observations), short observation period (tethered-balloon-based, aircraft-based, unmanned-aerial-vehicle-based observations), and low observation accuracy (lidar-based observation), making them less able to provide a complete and accurate $O_3$ vertical distribution for the whole lower troposphere, not to
mention $h$-reference lower-tropospheric $O_3$ climatology.

To our knowledge, ozonesonde represents the most accurate observation method for $O_3$ profiles in the troposphere. Therefore, in this study, we collected ozonesonde data observed in Beijing (northern China, Fig. 1a) and Hong Kong (southern China, Fig. 1a) to investigate the $h$-reference $O_3$ vertical distribution in the lower
troposphere over Chinese megacities. In addition, we also considered satellite-based $O_3$ precursor data, atmospheric composition reanalysis data, an integral method to determine the mixing layer top height $h$, and a photochemical indicator method to diagnose the $O_3$ production sensitivity. The specific aims of the study were to explore (1) the degree to which lower-tropospheric $O_3$ over megacities stratifies in the $h$-reference vertical coordinate system; (2) patterns in lower-tropospheric $O_3$ profiles in the $h$-reference vertical coordinate system; (3) how meteorological and
photochemical processes modulate $O_3$ vertical distribution patterns in the lower troposphere; and (4) differences in the characteristics and mechanisms of lower-tropospheric $O_3$ vertical distribution between Beijing and Hong Kong. These results of this study offer a reference for better understanding $O_3$ pollution in urban regions.

**2 Data and methods**



### 2.1 Ozonesonde measurements

We used ozonesonde data collected by the Beijing Nanjiao Observatory (116.47°E, 39.80°N, 33 m) and Hong Kong King's Park Observatory (114.17°E, 22.31°N, 66 m) from 2000 to 2022 (Fig. 1). Beijing Nanjiao Observatory is located in southern suburban of Beijing (Fig. 1b), while Hong Kong King's Park Observatory is situated within the urban core of Hong Kong (Fig. 1c). Both sites are affected by urban traffic emissions (Fig. 1b and c). Ozonesondes accompanied by radiosondes were regularly launched at approximately 13:30 local standard time (LST) once a week and provided high vertical resolution profiles of $O_3$, temperature, pressure, and humidity. We excluded data from ozonesondes launched outside a time window of 12:00–15:00 LST in order to minimizes changes in mixing layer $O_3$ arising from different launch times. We interpolated the original profiles on a fixed vertical grid of 20 m vertical resolution. To reduce uncertainties associated with data gaps, we further discarded (i) profiles with > 25% missing data between 0 and 4 km (i.e., accumulated data gaps of > 0.25 × 4,000 = 2,000 m), and/or (ii) profiles with > 10 missing data points between the surface and estimated mixing layer height (i.e., accumulated data gaps of > 10 × 20 = 200 m). After data exclusion, 1,897 ozonesondes were available for study: 924 soundings in Beijing and 973 soundings in Hong Kong. Figure 1d shows the monthly distribution of the available ozonesondes.

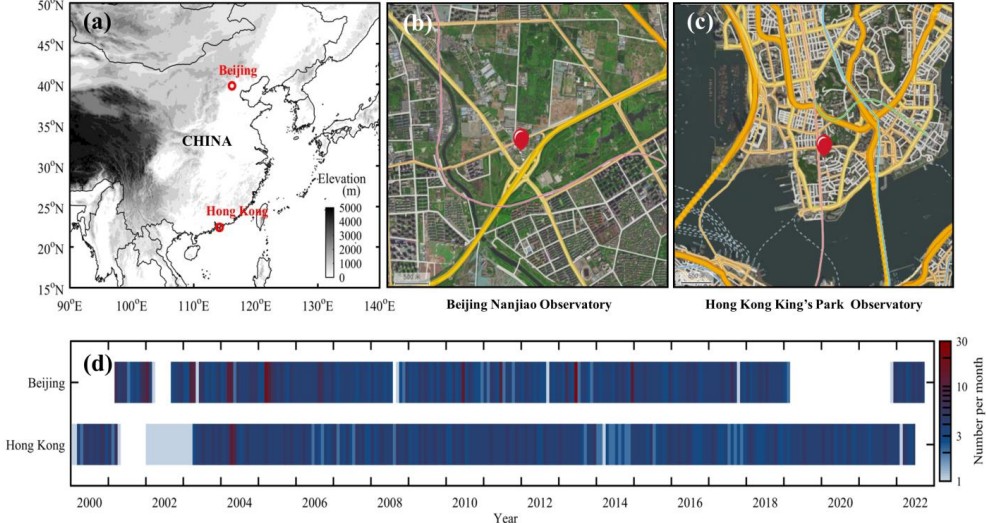

Figure 1. (a) Coordinates and surrounding environments of ozonesonde sites at (b) Beijing Nanjiao Observatory and (c) Hong Kong King's Park Observatory. (d) Monthly distribution of the available ozonesonde observations. Map image is © Amap.

### 2.2 Space-based ozone precursors

Level-3 formaldehyde ($CH_2O$, an indicator of VOCs) and nitrogen dioxide ($NO_2$, an indicator of NOx) column products from the Ozone Monitoring Instrument (OMI; https://disc.gsfc.nasa.gov/) were used to characterize $O_3$ precursor concentrations and diagnose $O_3$ production sensitivity. OMI is a nadir-looking near-UV/visible CCD spectrometer aboard the Aura satellite of NASA's Earth Observing System (Levelt et al., 2018). It provides global observations from 2004 onwards with a transit time at approximately 13:45 LST. Daily Level-3 products covering the period from 2004 to 2022 were adopted. The spatial resolution of the Level-3 $CH_2O$ products is $0.1° \times 0.1°$, and that of $NO_2$ is $0.25° \times 0.25°$; therefore, a bilinear interpolation method was used to resample OMI products to the



same resolution ($0.25^{o} \times 0.25^{o}$). We extracted daily data from a $3 \times 3$ grid region ($0.75^{o} \times 0.75^{o}$ centered on the ozonesonde site) and then averaged them to represent $O_3$ precursor columns for the respective sites.

### 2.3 Atmospheric composition reanalysis

Besides the OMI-based column products, pressure-level $CH_2O$, and $NO_2$ data from the fourth-generation European Center for Medium-Range Weather Forecasts (ECMWF) Atmospheric Composition Reanalysis (EAC4) were also used to characterize $O_3$ precursor concentrations and diagnose vertical $O_3$ production sensitivity. The EAC4 combines model data with global satellite observations into a complete and consistent dataset using a model of the atmosphere based on the laws of physics and chemistry (Inness et al., 2019). It was available at 3-h resolution for a horizontal resolution of $0.75^{o} \times 0.75^{o}$ and a vertical resolution of 7 layers below 700 hPa (1000, 950, 925, 900, 850, 800, and 700 hPa). EAC4 $CH_2O$ and $NO_2$ data at 06:00 UTC from 2003 onwards were used to support our interpretation of sonde-based $O_3$ vertical distribution.

### 2.4 Determination of mixing layer top height *h*

Several approaches have been developed to estimate *h* based on the gradient variation of individual atmospheric variables from radiosonde data (Seidel et al., 2010), including temperature (*T*), potential temperature (*θ*), relative humidity (*RH*), specific humidity (*q*), and atmospheric refractivity (*N*). However, there are substantial differences in the existing methods. Wang and Wang (2014) proposed a three-step method to integrate temperature, humidity, and cloud data to generate a consistent estimate of *h* from radiosonde profiles: Step 1, identify the height ($h_0$) that best meets the individual criteria for different atmospheric variables; step 2, derive the location of the cloud; and step 3, determine a consistent mixing layer height ($h_{con}$). We adopted this integral method to determine the mixing layer heights in Beijing and Hong Kong. Five atmospheric variables, namely, *T*, *θ*, *RH*, *q*, and *N*, were used. Among them, *T* and *RH* were measured by radiosonde, and the other variables were calculated from *T*, *RH,* and atmospheric pressure (Seidel et al., 2010). The upper limit of *h* was set to 4 km.

### 2.5 *h*-referenced vertical distribution and classification

Once the mixing layer height *h* was determined, all profiles were expressed in the *z/h* vertical coordinate system, where *z* is the actual altitude. In practice, atmospheric variables were interpolated along *z/h* values ranging between 0 (the surface) and 2 ($2 \times h$) with a vertical resolution of 0.05 (i.e., 41 altitude levels). For instance, if *h* on a specific profile was 1,000 m, the resampled profile extended from 0 to 2000 m with bins of 50 m. Hereafter, this type of vertical profile is denominated as a mixing-layer-height-referenced (i.e., *h*-referenced) profile. In this *z/h* vertical coordinate system, mixing-layer $O_3$ was denominated as $MLO_3$ and free-tropospheric $O_3$ was denominated as $FTO_3$. Based on Student's *t* test, we further classified individual *h*-referenced $O_3$ profiles into three distinct patterns: $MLO_3$-dominated (mean $MLO_3$ significantly higher than mean $FTO_3$ at a significance level of 0.01); $FTO_3$-dominated (mean $MLO_3$ significantly lower than mean $FTO_3$ at a significance level of 0.01); and uniform distribution (no significant differences between the means of $MLO_3$ and $FTO_3$).

### 2.6 Diagnosis of ozone production sensitivity

$O_3$ is photochemically generated when its precursors (e.g., NOx and VOCs) are abundant in the presence of sunlight (Seinfeld and Pandis, 2016). Owing to complex chemical mechanisms and regional differences in emissions and meteorology, the relationship between $O_3$ and its precursors involves highly non-linear interactions (Jin et al., 2020). Under high VOC and low NOx conditions, $O_3$ production is not sensitive to VOCs, but is positively correlated to NOx (i.e., a NOx-sensitive regime). Under low VOC and high NOx conditions, $O_3$ production tends to increase with VOC growth or NOx reduction (i.e., VOC-sensitive regime). In this study, the





CH$_2$O/NO$_2$ ratio (FNR) was used as the photochemical indicator to diagnose O$_3$ production sensitivity. An inherent challenge of this diagnosis approach is that FNR thresholds marking the VOC–NOx transition regime are likely distinct from region to region (Jin et al., 2020). For the NCP region (including Beijing), Li et al. (2021) diagnosed

the transition regime as occurring when FNR ranges from 1.2 to 2.1; for the PRD region (including Hong Kong), Liao et al. (2021) diagnosed the transition regime as occurring when FNR ranges from 1.0 to 1.5. Ratios below and above these ranges indicate VOC-limited O$_3$ production and NOx-limited regimes, respectively. These localized FNR thresholds were adopted in this study to diagnose O$_3$ production sensitivity.

**3 Results and discussion**

**3.1 Lower-tropospheric ozone climatology**

Figure 2 shows the traditional lower tropospheric O$_3$ climatology of Beijing and Hong Kong. Seasonal results are averaged from ozonesonde profiles collected in spring (M–A–M), summer (J–J–A), autumn (S–O–N), and winter (D–J–F). There is a typical summer-high-winter-low seasonality in lower tropospheric O$_3$ over Beijing, with the

highest O$_3$ concentrations in June. Such seasonality is broadly similar to previous tropospheric O$_3$ climatology based on lesser O$_3$ profiles in Beijing (Ding et al., 2008;Zhang et al., 2021). In photochemical active months (May–August), high-concentration O$_3$ is photochemically produced throughout the lower troposphere, particularly in the mixing layer, causing an isolated O$_3$-peak area (> 100 ppbv) near the upper mixing layer. In other months, strong urban NO-titration accompanied by weak O$_3$ production causes a positive vertical gradient of O$_3$ concentration in

the lower troposphere; the average vertical gradient of O$_3$ reaches a maximum in winter.

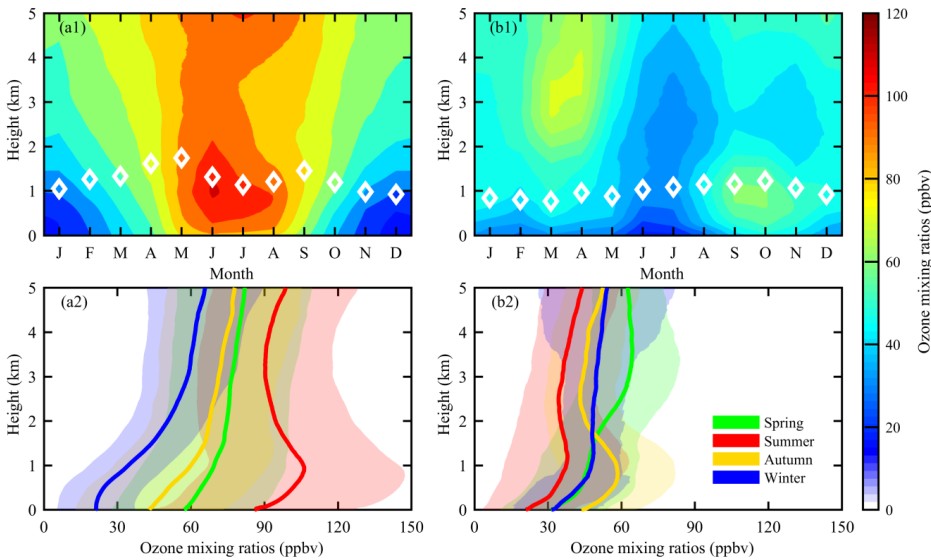

Figure 2. Lower tropospheric ozone vertical distribution over (a) Beijing and (b) Hong Kong. (1) Monthly variation. (2) Seasonal variation. The white diamonds in (1) represent the monthly mean mixing layer height.

Lower tropospheric O$_3$ climatology in Hong Kong is remarkably different from that in Beijing. In particular, lower tropospheric O$_3$ is low in the summertime (< 40 ppbv). Similar O$_3$ minima have been reported in other subtropical cities in Eastern Asia, such as Hanoi and Naha (Liao et al., 2021;Oltmans et al., 2004;Ogino et al., 2013) and likely reflect the influence of the Asian summer monsoons, which bring maritime air with low O$_3$ northward




from the tropical Pacific to subtropical regions. Although Beijing is also impacted by the Asian summer monsoons,
these ocean-sourced air masses become enriched with $O_3$ precursors while passing over polluted eastern China,
leading to an accumulation of $O_3$ over Beijing. Interestingly, there are two isolated areas of $O_3$ enhancement over
Hong Kong, those in the lower free troposphere (~3.5 km) from March to April and in the upper mixing layer (~0.8
km) in autumn. The former is attributed to long-range transport of wildfire-related $O_3$ production in the upwind
Indochina Peninsula; the latter results from local $O_3$ production via photochemical reactions under hot and dry
weather conditions in autumn (Liao et al., 2021).

### 3.2 Mixing-layer-height-referenced ozone vertical distribution

We investigated the climatological vertical stratification of $O_3$ below and above the ML–FT interface (i.e.,
mixing layer height $h$) over Beijing and Hong Kong (Fig. 3). This $h$-referenced $O_3$ climatology provides an
additional dimension (further categorization by mixing layer height) not available in traditional vertical ozone
profile climatology. The significant disparity between the $h$-referenced $O_3$ climatology (Fig. 3) and traditional $O_3$
climatology (Fig. 2) illustrates how much information is lost using simple ozonesonde averages. For example, the
$h$-referenced $O_3$ profiles show a clear inflexion (or discontinuity) at the interface between the ML and FT ($z/h$=1),
which is not apparent in the traditional $O_3$ climatology. These results reflect the fact that mixing-layer capping
inversion acts as an effective although porous geophysical barrier that limits vertical exchange between the ML and
FT, leading to distinct $O_3$ levels on either side.

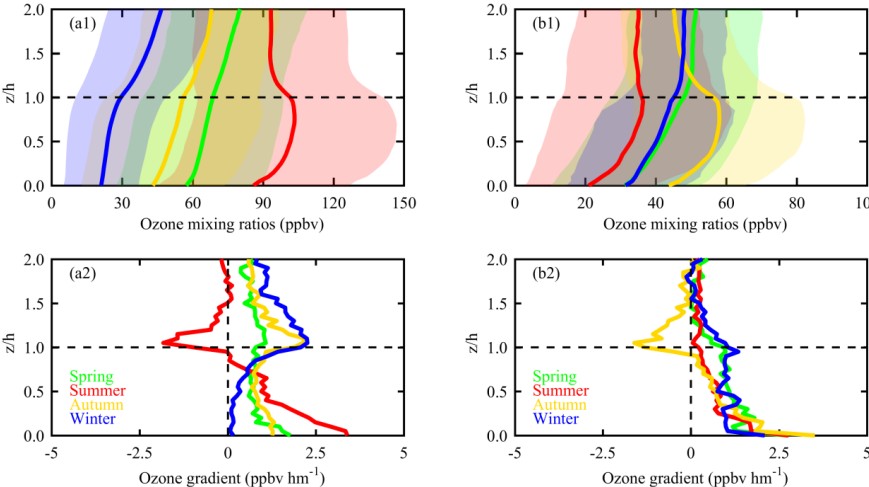

Figure 3. Mixing-layer-height-referenced ozone vertical distribution over (a) Beijing and (b) Hong Kong. (1) Ozone
mixing ratio profile. (2) Ozone gradient profile.


In Beijing, seasonal $O_3$ profiles in autumn, winter, and spring present a low-ML/high-FT vertical distribution
pattern with $O_3$ mixing ratios that increase with altitude throughout the lower troposphere and variable vertical
gradients depending on season and altitude. Generally, the strongest gradients are observed either close to the
surface or near the ML–FT interface. Near the surface, they are likely due to strong $O_3$ titration by NO emitted from
urban traffic (Karl et al., 2023). Near the ML–FT interface, they are likely attributable to the barrier effect of
mixing-layer capping inversion. The $O_3$ gradients gradually decrease with altitude above the ML–FT interface;
below the interface, they slightly decrease with altitude in spring and autumn but gradually increase with altitude in





winter. Winter $O_3$ gradients are almost zero in the surface layer ($z/h < 0.4$), reflecting strong titration that often causes $O_3$ to be almost completely depleted in the lower ML. In summer, the averaged $O_3$ profile exhibits a sickle-shape pattern, with a marked drop in concentrations from the upper ML to the lower FT. Summer $O_3$ gradients quickly decrease with altitude inside the ML and eventually become negative near the ML–FT interface. The maximum negative gradient ($-2.2$ ppbv hm$^{-1}$) occurs just above the mixing layer top height. In Hong Kong, the averaged $O_3$ profiles in winter and spring present low-ML/high-FT vertical distribution, similar to Beijing. However, the autumn averaged $O_3$ profile shows a sickle-shape pattern, similar to the summer profile in Beijing. In contrast, the summer averaged $O_3$ profile in Hong Kong displays a transitional feature from spring to autumn, characterized by a weak $O_3$ peak just below the ML–FT interface. Compared with Beijing, the $O_3$ gradients in Hong Kong vary across a smaller range; however, they are commonly sharper in the surface layer.

For both Beijing and Hong Kong, the highly variable $O_3$ gradients in the ML confirm that the well-mixed ML remains a large exception for $O_3$, even on summer afternoons when vertical turbulent mixing is expected to be strongest. In particular, the increasing $O_3$ with altitude in the lower ML indicates that strong photochemistry and vertical mixing on summer afternoons is insufficient to quickly compensate for $O_3$ titration consumption ($NO+O_3 \rightarrow NO_2$) in the surface layer, where NO is largely emitted by urban traffic. A previous study indicated that $MLO_3$ evolution in urban areas adheres to vertical physiochemical circulation involving multiple reactions in the $O_3$–NO–$NO_2$ triad (Tang et al., 2017). NO emissions react with $O_3$ to generate $NO_2$ near the ground, which is then transported vertically to the upper ML; $O_3$ is generated by $NO_2$ photolysis in the upper ML and is then transported down to the surface layer to compensate for the loss by NO titration. In this process, the titration process is thought to drive the downward flux of $O_3$ into the urban roughness layer (Karl et al., 2023). Under favorable weather conditions, high-concentration $MLO_3$ production can greatly modify the vertical profile of $O_3$ from the more customary low-ML/high-FT vertical distribution to a high-ML/low-FT vertical distribution. This modification is thought to be episodic in low-emission cities (e.g., Frankfurt; (Petetin et al., 2016); in such cities, the vertical structure of averaged $O_3$ profiles in the photochemical active season (e.g., summer) remains low-ML/high-FT the same throughout the year (Petetin et al., 2018). However, in high-emission megacities (e.g., Beijing and Hong Kong), photochemistry-driven modification can be expected to be common during in the photochemical active season (summer in Beijing and autumn in Hong Kong), eventually causing a seasonal sickle-shape $O_3$ profile in the lower troposphere. These seasonal differences in lower tropospheric $O_3$ profiles imply that the aforementioned transport barrier to vertical exchange has different connotations, typically changing from a ML-to-FT detrainment barrier in summer (autumn) to a FT-to-ML entrainment barrier in other seasons in Beijing (Hong Kong).

The vertical autocorrelation of $O_3$ in the $z/h$ vertical coordinate system was further analyzed to investigate the links between the ML and FT. Based on all individual $O_3$ profiles, we calculated the correlation coefficients of $O_3$ between the different pairs of $z/h$ altitude levels. The obtained $O_3$ vertical autocorrelation matrix is shown in Figure 4. Within both the ML ($z/h$ between 0 and 1) and FT ($z/h$ between 1 and 2), we found strong correlations (usually > 0.90, mean of 0.97 in Beijing; > 0.85, mean of 0.91 in Hong Kong). However, the correlations between the two atmospheric compartments (ML vs. FT) decreased quickly with vertical distance, with means of 0.84 in Beijing and 0.60 in Hong Kong. In general, correlations in Hong Kong were found to be weaker than those in Beijing. This can be explained by two possible reasons. (i) Hong Kong is a coastal city, where clean maritime air and polluted continental air can dominate at different altitudes (e.g., sea-land breeze); therefore, distinct air mass sources can weaken the correlation of $O_3$ between different altitude levels. (ii) Hong Kong is located in humid zone, where surface sensible heat is relatively weaker than that in semi-humid zones (e.g., Beijing); therefore, weak turbulent convection causes weak mixing of $O_3$ in the vertical direction (Xu et al., 2021). The iso-correlation contours in both





megacities present a "W" shape along the diagonal direction, with the inflexion point at z/h = 1. This is consistent with northern mid-latitude findings in Petetin et al. (2018), indicating that stratification occurs most commonly at the ML–FT interface.

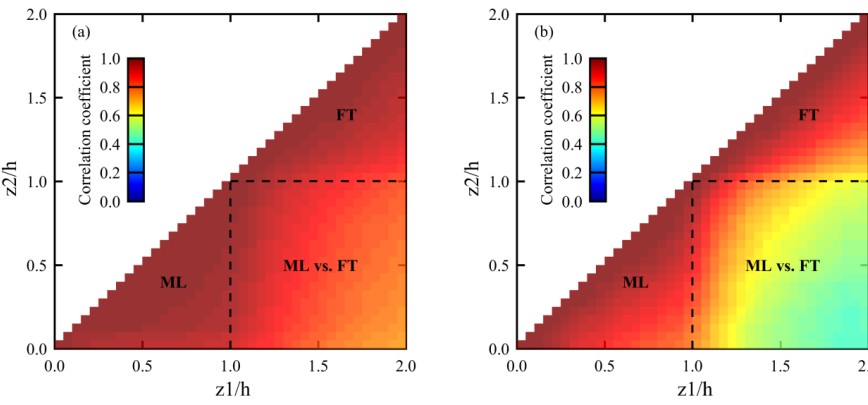

Figure 4. Auto-correlation of ozone mixing ratios between different z/h altitude levels over (a) Beijing and (b) Hong Kong. Dashed lines separate three areas involving correlation within the mixing layer (ML), within the free troposphere (FT), and between the mixing layer and free troposphere (ML vs. FT).

Both surface concentration and vertical distribution of $O_3$ are highly variable at the synoptic scale and can greatly depart from standard climatology depending on meteorological conditions and the availability of $O_3$ precursors. Based on Student's $t$ test, all individual $h$-referenced $O_3$ profiles were further classified into three typical patterns to investigate synoptic climatology of lower tropospheric $O_3$ in Beijing and Hong Kong. The statistical results indicate that the $FTO_3$-dominated pattern occurs most frequently in both megacities. The respective occurrence frequencies of $FTO_3$-dominated, uniform, and $MLO_3$-dominated distributions were 69%, 11%, and 20% in Beijing, and 54%, 21%, and 25% in Hong Kong, respectively. Figure 5 shows the composite of $O_3$ (gradient) profiles according to the different $O_3$ profile patterns in Beijing and Hong Kong. In the $FTO_3$-dominated pattern, averaged $FTO_3$ concentrations are 61.6 ppbv in Beijing and 44.9 ppbv in Hong Kong, which are 15 and 13 ppbv higher than the averaged $MLO_3$ concentrations in the respective cities. Such concentration differences between $FTO_3$ and $MLO_3$ cause a sharp positive gradient of $O_3$ near the ML–FT interface (2.3 ppbv hm$^{-1}$ in Beijing and 1.8 ppbv hm$^{-1}$ in Hong Kong). For the $MLO_3$-dominated pattern, averaged $MLO_3$ concentrations are 109.8 ppbv in Beijing and 62.2 ppbv in Hong Kong, ~18 ppbv higher than the averaged $FTO_3$ concentrations in both cities, causing a steep negative gradient of $O_3$ near the ML–FT interface (−4.3 ppbv hm$^{-1}$ in Beijing and −3.8 ppbv hm$^{-1}$ in Hong Kong). For the uniform distribution, despite no significant difference in the means of $MLO_3$ and $FTO_3$, the composited $O_3$ profile shows an "S" shape pattern with a slightly negative gradient (approximately −1.0 ppbv hm$^{-1}$) near the ML–FT interface.





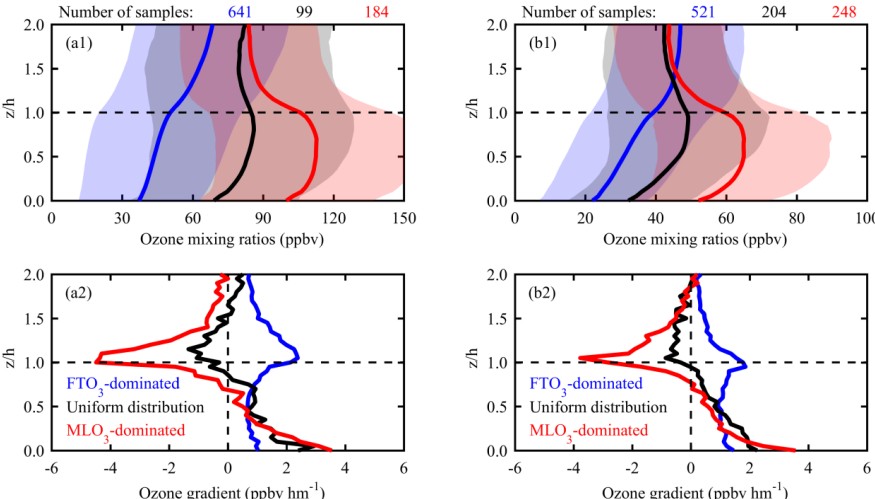

Figure 5. Composites of (1) *h*-referenced ozone profiles and (2) *h*-referenced ozone gradient profiles according to
330    different patterns in (a) Beijing and (b) Hong Kong.

Figure 6 shows occurrence frequencies of the three distinct $O_3$ profile patterns in different seasons and mixing
layer height bins. In Beijing, while the $FTO_3$-dominated pattern prevails in winter (94.2%), autumn (79.1%), and
spring (75.3%), the $MLO_3$-dominated pattern prevails in summer (46.3%). In Hong Kong, the $FTO_3$-dominated
335    pattern occurs frequently in spring (67.7%), winter (65.8%), and summer (55.8%), and the $MLO_3$-dominated
pattern prevails in autumn (55.1%). Such frequent occurrence of $MLO_3$-dominated patterns confirms our theory
that the $MLO_3$-dominated pattern is common rather than episodic in the photochemical active season of
high-emission Chinese megacities. In contrast, the occurrence dependence of $O_3$ profile patterns on mixing layer
height is not as strong as that on season. The $FTO_3$-dominated pattern prevails in most *h* bins, particularly in
340    Beijing. Nevertheless, the $MLO_3$-dominated pattern is still relatively more frequent in the *h* bin between 1.0 and 2.0
km (27.3% in Beijing and 36.7% in Hong Kong) than in lower and higher *h* bins. This is to some degree consistent
with the findings of Zhao et al. (2019), who revealed that moderate mixing layer height is usually accompanied by
very favorable meteorological (moderate RH and high temperature) and photochemical ($NO_x$–VOC transition
regime) conditions for high-concentration $MLO_3$ production.

345





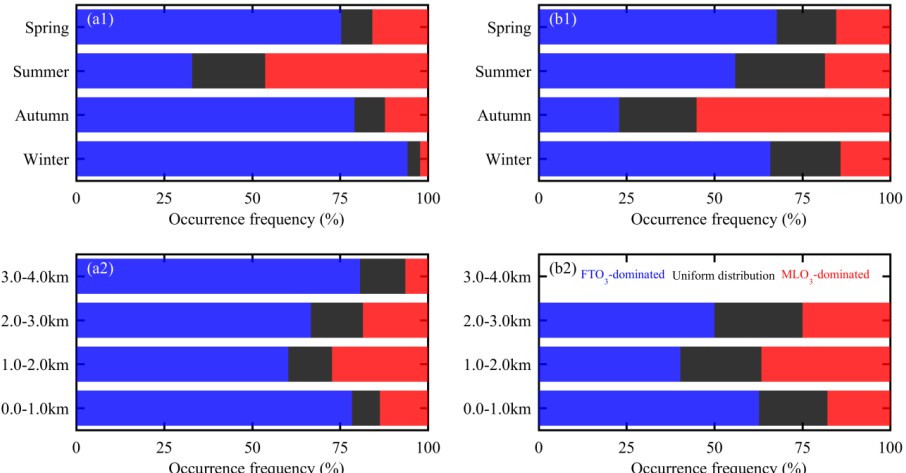

Figure 6. Occurrence frequencies of three *h*-referenced ozone profile patterns according to (1) season and (2) mixing layer height bins in (a) Beijing and (b) Hong Kong (in Hong Kong, no case is found for *h* > 3.0 km).

### 3.3 Mechanistic understanding of distinct ozone profile patterns

#### 3.3.1 Meteorological interpretations

The *h*-referenced profiles of potential temperature ($\theta$), relative humidity (RH), and wind speed (WS) according to different $O_3$ profile patterns in Beijing and Hong Kong are shown in Figure 7. Near the surface (z/h < 0.1), potential temperature decreases with altitude in both megacities, indicating a shallow superadiabatic layer due to daytime surface radiation heating. As expected, in other parts of the ML, potential temperature profiles in Beijing are neutral adiabatic within the afternoon convective ML (Stull, 1988). However, the corresponding profiles in Hong Kong are subadiabatic, implying insufficient thermal convection mixing over this coastal city, which may partly explain the lower autocorrelations of $O_3$ between different altitude levels within the ML in Hong Kong (Fig. 4). In the FT (z/h > 1.0), potential temperature increases with altitude with a relatively larger gradient than that in the ML. As expected, there is a sharp increase in potential temperature at the ML–FT interface where positive vertical gradients reach 1.0 $^o$C hm$^{-1}$ on average. This maximum gradient is indicative of strong mixing layer capping inversion. However, the maximum gradient values are almost identical among the different $O_3$ profile patterns, suggesting that capping inversion acts as a transport barrier to suppress $O_3$ vertical exchange but is not responsible for the different directions of vertical exchange (i.e., FT-to-ML entrainment or ML-to-FT detrainment). In fact, no structural change was found in the averaged $\theta$ profiles among the different patterns. Similar to the $\theta$ profiles, the RH and WS profiles shared an analogous vertical structure among different $O_3$ profile patterns in both megacities. In general, RH and WS levels inside the ML were higher and lower, respectively, than the corresponding lower troposphere levels.

Without considering the vertical structure, values of aforementioned meteorological variables differed among the $O_3$ profile patterns, suggesting that meteorological conditions are the main regulating factors of lower troposphere $O_3$ levels. From the FTO$_3$-dominated to MLO$_3$-dominated pattern, potential temperature increases in both megacities. The high temperature of the MLO$_3$-dominated pattern favors high-concentration $O_3$ production in the ML. The cross-pattern value change of RH and WS shows some differences between Beijing and Hong Kong. For example, on MLO$_3$-dominated days, RH is moderate in Beijing but low in Hong Kong, and when WS is low in Beijing it is moderate in Hong Kong. Nevertheless, RH and WS inside the ML of Hong Kong are always higher





than those in the ML of Beijing. While humid air tends to suppress photochemical reactions, windy condition tends to inhibit ozone accumulation. The higher RH and WS conditions may partly explain the lower $O_3$ levels in Hong Kong. From above analyses a key factor leading to the $MLO_3$-dominated pattern in both megacities is high temperature. Previous studies have indicated that high temperature not only increases the $O_3$ production rate (Wang et al., 2022a), but also strengthens the volatilization rate of $O_3$ precursors, particularly biomass VOC emissions (Duncan et al., 2009).

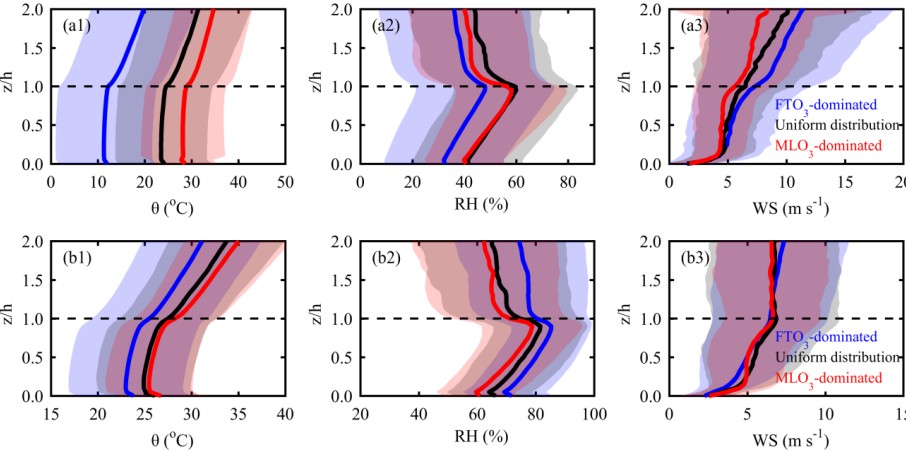

Figure 7. Composited $h$-referenced profiles of (1) potential temperature, (2) relative humidity, and (3) wind speed according to different ozone profile patterns in (a) Beijing and (b) Hong Kong.

### 3.3.2 Photochemical interpretations

Figure 8 shows composited column concentrations and vertical distributions of $CH_2O$ and $NO_2$ according to the different $O_3$ profile patterns. In Beijing, tropospheric $CH_2O$ columns are 7.7 ($\pm$4.3) $\times10^{15}$, 11.8 ($\pm$6.9) $\times10^{15}$, and 12.4 ($\pm$5.7) $\times10^{15}$ molec. cm$^{-2}$ in the $FTO_3$-dominated, uniform distribution, and $MLO_3$-dominated patterns, respectively; the corresponding values for the tropospheric $NO_2$ column are 19.2 ($\pm$10.7) $\times10^{15}$, 14.5 ($\pm$9.8) $\times10^{15}$, and 13.2 ($\pm$7.7) $\times10^{15}$ molec. cm$^{-2}$, respectively. From the $FTO_3$-dominated pattern to the $MLO_3$-dominated pattern, $CH_2O$ increases throughout the lower troposphere (up to 700 hPa) with a maximum increment in the upper ML (~900 hPa). This maximum increment can be explained by high-elevation biogenic VOC emissions in the western mountains (i.e., Taihang Mountains) and vertical mixing of VOC emissions in the southern NCP during the transport process. In contrast, $NO_2$ mainly decreases in the lower ML (below 900 hPa), especially in the surface layer (1000 hPa). The cross-pattern change of $O_3$ precursors is likely attributable to significant seasonality of precursor emissions in Beijing and the NCP (e.g., anthropogenic emissions change between heating and non-heating periods and natural emissions change between leafy and leafless periods). As shown in Figure 6, the $MLO_3$-dominated pattern prevails in the warm season (Spring–Autumn), during which heating-related precursor emissions (mainly $NO_x$) are negligible but biogenic precursor emissions (mainly VOCs) are considerable (Fig. 7). This explains the elevated $CH_2O$ column and low $NO_2$ column in the $MLO_3$-dominated pattern in Beijing.

In Hong Kong, both $CH_2O$ and $NO_2$ had lower concentrations than those in Beijing (except for $CH_2O$ in the $FTO_3$-dominated pattern). This partly explains the lower $O_3$ levels in Hong Kong. The cross-pattern change of $O_3$ precursors (regardless of total column or vertical distribution) in Hong Kong follows the same order as that in Beijing (i.e., increase of $CH_2O$ and decrease of $NO_2$ from $FTO_3$-dominated to $MLO_3$-dominated). In Hong Kong,



tropospheric $CH_2O$ columns are 8.5 ($\pm$ 4.8) $\times 10^{15}$, 8.5 ($\pm$ 3.8) $\times 10^{15}$, and 9.1 ($\pm$ 4.1) $\times 10^{15}$ molec. $cm^{-2}$ in the $FTO_3$-dominated, uniform distribution, and $MLO_3$-dominated patterns, respectively. The corresponding values for tropospheric $NO_2$ columns are 11.8 ($\pm$ 6.8) $\times 10^{15}$, 11.6 ($\pm$ 6.8) $\times 10^{15}$, and 11.0 ($\pm$ 7.6) $\times 10^{15}$ molec. $cm^{-2}$. Evidently,

cross-pattern change amplitudes of $O_3$ precursors in Hong Kong are smaller than those in Beijing. In subtropical Hong Kong and the PRD region, $O_3$ precursor emissions are not affected by heating-related emission changes from warm to cold season, and are less affected by seasonal biogenic emission changes because of evergreen leaves throughout the year. This weak seasonal dependence of precursor emission leads to small cross-pattern differences in $CH_2O$ and $NO_2$ in Hong Kong. The slight differences in $O_3$ precursors among the different $O_3$ profile patterns in

Hong Kong are likely attributable to temperature-driven precursor changes. Higher temperatures in the $MLO_3$-dominated regime trigger the release of VOC emissions, causing higher $CH_2O$ concentrations, and photolysis of $NO_2$, causing lower $NO_2$ concentrations.

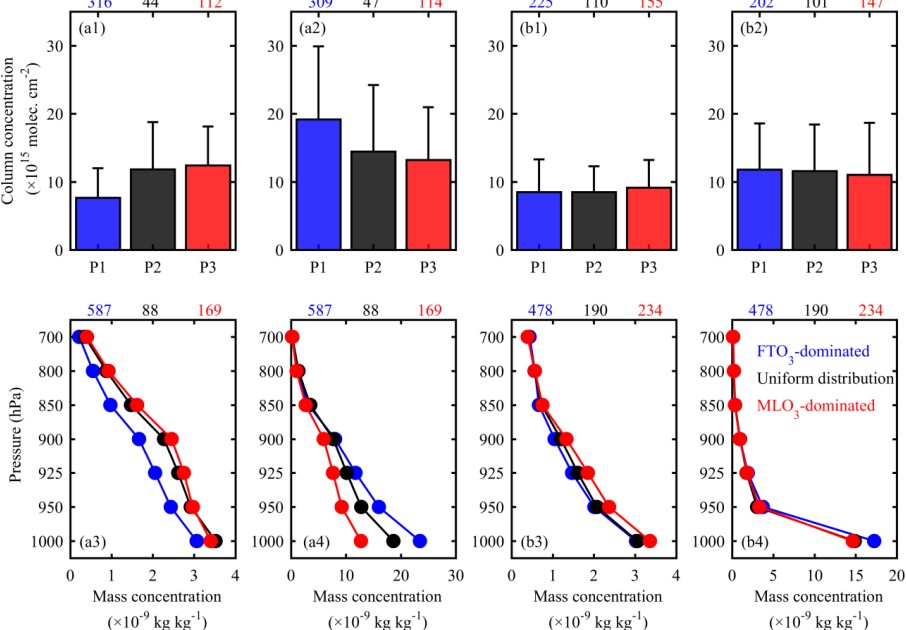

Figure 8. Composite column concentrations (upper panel) and vertical distributions (lower panel) of ozone

precursors according to different ozone profile patterns in (a) Beijing and (b) Hong Kong. $CH_2O$ (1 and 3); $NO_2$ (2 and 4). P1: $FTO_3$-dominated pattern; P2: Uniform distribution pattern; P3: $MLO_3$-dominated pattern.

$O_3$ production chemistry in urban areas is usually VOC-limited (Li et al., 2021), and we found that $CH_2O$ concentrations in both Beijing and Hong Kong increased from the $FTO_3$-dominated to the $MLO_3$-dominated pattern,

suggesting increased potentiality of high-concentration $O_3$ production. This potentiality can be easily realized on the $MLO_3$-dominated days owing to the hot-dry weather conditions, providing favorable meteorology for photochemical reactions. That is to say, the $MLO_3$-dominated pattern in Beijing and Hong Kong is driven by intensive photochemical production of $MLO_3$ under elevated VOC and high temperature conditions. Since the increment of $CH_2O$ is larger in the upper mixing layer, higher-concentration $O_3$ production can be expected in

upper mixing layer, in consistent with the observational result of sickle-shape $O_3$ profile in the lower troposphere. Conversely, the $FTO_3$-dominated pattern is likely due to strong titration consumption of $MLO_3$ under elevated $NO_2$





and low temperature conditions. Owing to the nonlinear relationship between $O_3$ and its precursors (i.e., VOC and NOx), net production of $O_3$ is subject to both absolute concentrations of VOC and NOx and their relative ratio, which determines the $O_3$ production sensitivity. Based on the FNR photochemical indicator method, we further
diagnosed $O_3$ production sensitivity to examine the potential change in $O_3$ production chemistry among different $O_3$ profile patterns.

Figure 9 shows the scatter distribution of OMI-based tropospheric $CH_2O$ and $NO_2$ columns over Beijing and Hong Kong. In Beijing, most (~90%) points associated with the $FTO_3$-dominated pattern are located in the
VOC-limited regime; these points correspond to high $NO_2$ concentrations, indicating large $O_3$ consumption via the titration reaction ($NO+O_3 \rightarrow NO_2$). Strong titration can therefore partly explain the low $MLO_3$ concentrations in the $FTO_3$-dominated pattern (another explanation is weak $O_3$ production due to low temperature). In other patterns, ~27% of days are identified as being in the transition regime, which usually represents optimal VOC-NOx ratios for net production of $O_3$. High-efficiency $O_3$ production likely offsets NO-titration $O_3$ consumption, leading to higher
$MLO_3$ concentration compared with that in the $FTO_3$-dominated pattern. In contrast, in Hong Kong, points that belong to different $O_3$ profile patterns mix well together, forming similar frequency matrices of $O_3$ production sensitivity among the different patterns. Compared with the rare occurrence of the $NO_x$-limited regime in Beijing, the NOx-limited regime is more common in Hong Kong (> 20%), indicating that ozone production chemistry is more sensitive to $NO_x$ in Hong Kong than in Beijing. Since the partly integral concentration of $O_3$ precursors in the
lower layer holds a significantly higher proportion of the total column concentration, the OMI-based analyses are more likely to represent near-surface characteristics of $O_3$ production sensitivity, particularly in Hong Kong (Fig. 8). Therefore, it is necessary to further explore the vertical characteristics of $O_3$ production sensitivity.

Figure 10 shows the EAC4-based vertical characteristics of $O_3$ production sensitivity. In Beijing, the averaged
FNR values in the ML (< 850 hPa) differ significantly among the different $O_3$ profile patterns. However, all are located in the VOC-limited regime. From 850 to 700 hPa, the averaged FNR values increase quickly, causing a shift of $O_3$ production sensitivity from VOC-limited to NOx-limited. There is a significant increase of the transition from $FTO_3$-dominated to $MLO_3$-dominated, whereas the occurrence of the VOC-limited regime shows an opposite trend. Vertically, the transition regime frequency increases with height in the ML (< 850 hPa), regardless of the $O_3$ profile
pattern. This is broadly similar to the MAX–DOAS-based findings of Chi et al. (2018), who reported that the transition regime accounted for 27.3% at 300 m height, but 50.0% at 1,100 m height over Beijing. In Hong Kong, FNR values increase more rapidly with height than in Beijing, but show small differences among the different $O_3$ profile patterns. The shift height of $O_3$ production sensitivity in Hong Kong is lower than that in Beijing. Based on averaged FNR profiles, the shift in Hong Kong occurs in the height range between 950 and 925 hPa. The daily
occurrence statistics also reveal a higher transition regime frequency in this height range. Below this height, $O_3$ production chemistry is overwhelmingly controlled by the VOC-limited regime, and above by the NOx-limited regime. Similar results had been reported via MAX–DOAS observations in Guangzhou (a megacity ~110 km northwest of Hong Kong), where $O_3$ production sensitivity changed with height from VOC-limited (0.02–0.22 km) to transitional (0.22–0.42 km) to NOx-limited (0.42–2.02 km) (Lin et al., 2022).

The above results demonstrate that the ozone precursor level and ozone production sensitivity play an important role in modulating the vertical distribution of $O_3$ in the lower troposphere. To be specific, the changes of ozone precursor level and ozone production sensitivity determine the final chemical behavior of $O_3$ (destruction or production) in the mixing layer among the different $O_3$ profile patterns. In the $FTO_3$-dominated pattern, the
supersaturated $NO_x$ concentrations trigger significant $MLO_3$ destruction under the overwhelming VOC-limited



regime condition, causing a significantly lower $O_3$ in the mixing layer than that in the free troposphere. Therefore, the mixing layer capping inversion acts as a barrier for FT-to-ML $O_3$ entrainment in the FTO$_3$-dmoniated pattern. From the FTO$_3$-dominated pattern to MLO$_3$-dominated pattern, the increased $CH_2O$ concentrations push $O_3$ production sensitivity away from the VOC-limited regime (towards higher $NO_x$ sensitivity) and favor for ozone photochemically production. The net production of $O_3$ is expected to be larger in the upper mixing layer, where larger increase of $CH_2O$ occurs. As a result, the MLO$_3$-dominated pattern is expressed as a sickle-shape $O_3$ profile (the highest $O_3$ level in the upper mixing layer), reflecting a ML-to-FT $O_3$ detrainment barrier effect of mixing layer capping inversion.

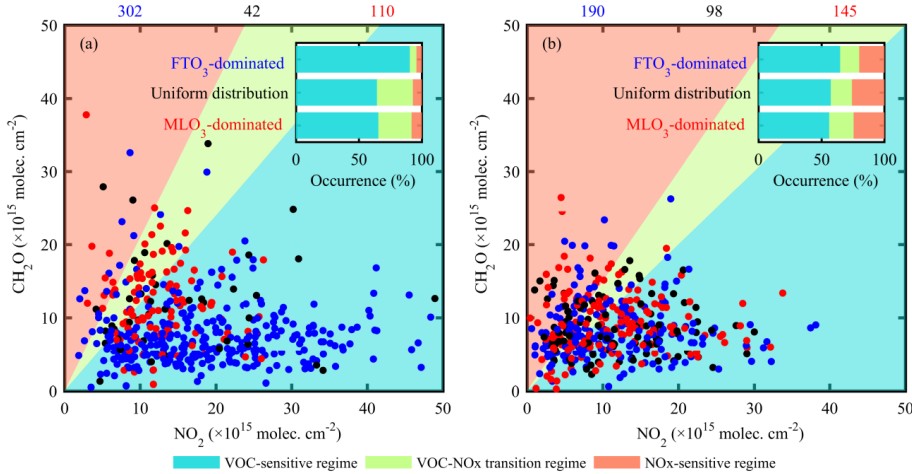

Figure 9. Ozone production sensitivity of different ozone profile patterns in (a) Beijing and (b) Hong Kong. Scatter distribution of $CH_2O$ and $NO_2$ in different ozone production sensitivity regimes (1 and 3); statistical occurrence of daily ozone production sensitivity (2 and 4).

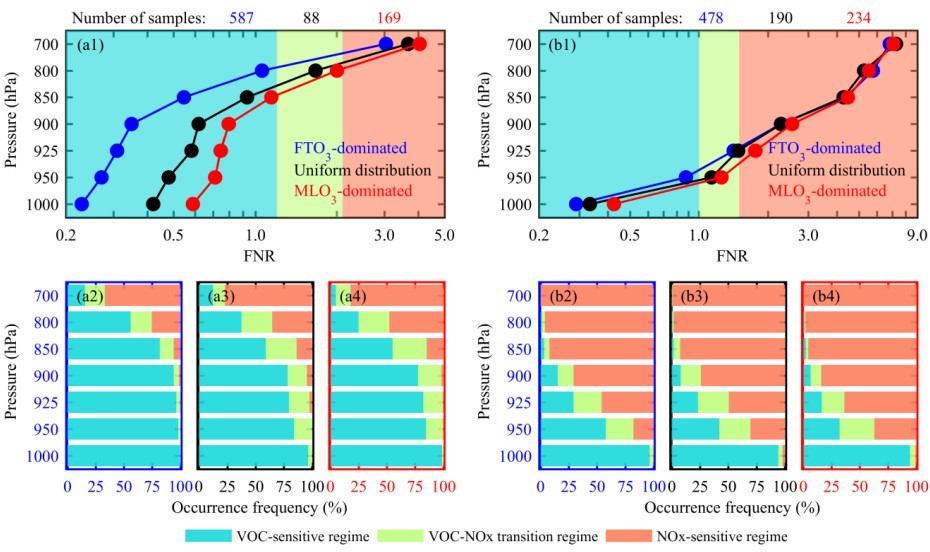



Figure 10. Vertical characteristics of ozone production sensitivity according to different ozone profile patterns in (a) Beijing and (b) Hong Kong. Upper panels (1) denote vertical FNR profiles. Lower panels (2, 3, and 4) denote the occurrence frequency of ozone production sensitivity in $FTO_3$-dominated, uniform distribution, and $MLO_3$-dominated patterns.


## 4 Summary

We investigate lower tropospheric $O_3$ distribution over two Chinese megacities (Beijing and Hong Kong) by introducing a novel mixing-layer-height-referenced ($h$-referenced) $O_3$ climatology, in which lower tropospheric $O_3$ profiles are scaled according to the mixing layer top heights. Mixing layer top height was determined by an integral method that integrates temperature, humidity, and cloud profiles. We focused on the lower troposphere (below $2\times$ the mixing layer top height), with each profile subdivided in two compartments: the mixing layer and free troposphere (ML and FT). By examining $O_3$ concentration differences between the ML and FT (i.e., $MLO_3$ and $FTO_3$), all individual $O_3$ profiles were classified into three typical patterns: $MLO_3$-dominated, $FTO_3$-dominated, and uniform distribution. Sonde-based meteorological profiles and multi-source $O_3$ precursors ($CH_2O$ and $NO_2$) were further analyzed to characterize the main physiochemical processes driving contrasting $O_3$ budgets among the different $O_3$ profile patterns. Our conclusions are as follows:

(1) Compared with traditional sea-level-referenced climatology, $h$-referenced $O_3$ climatology preserves the dependence of $O_3$ abundance and its variability on mixing layer top height, highlighting an inflexion point (or discontinuity) at the interface between the ML and FT.

(2) Lower tropospheric $O_3$ concentrations show summer-high/winter-low climatology in Beijing, and autumn-high/summer-low climatology in Hong Kong. In the photochemical active season (summer in Beijing and autumn in Hong Kong), seasonal $O_3$ profiles exhibit a sickle-shape pattern with a marked drop in concentrations from high values in the upper ML to low values in the lower FT. This sickle-shape profile pattern is significantly different from monotone increasing profile patterns across the rest of the year.

(3) Highly variable $O_3$ gradients in the lower troposphere, particularly at the surface layer and ML–FT interface, reflect the universality of vertical $O_3$ stratification structure. $O_3$ stratification in Hong Kong is stronger than that in Beijing. The stratification in the surface layer is likely due to strong titration chemical processes, and that at the ML–FT interface is attributable to the dynamic transport barrier of mixing layer capping inversion on vertical exchange. The contrasting $O_3$ gradients at the ML–FT interface indicate different transport barrier effects, which typically shift from a ML-to-FT detrainment barrier in summer (autumn) to a FT-to-ML entrainment barrier in other seasons in Beijing (Hong Kong).

(4) $FTO_3$-dominated pattern represents the most common $O_3$ profile patterns in both Beijing and Hong Kong (occurrence frequencies of 69% and 54%, respectively). However, $MLO_3$-dominated pattern prevails in the photochemical active season, accounting for 46% of summer days in Beijing and 55% of autumn days in Hong Kong, which is more frequent than the previously reported episodic occurrence in northern mid-altitudes, indicating intensive $MLO_3$ production in high-emission Chinese megacities.

(5) There are no structural differences in lower tropospheric meteorological profiles (θ, RH, and WS) among the different $O_3$ profile patterns. The maximum positive θ gradient at the ML–FT interface demonstrates the common existence of mixing layer capping inversion, which acts as a barrier to vertical exchange. In the $FTO_3$-dominated pattern, $MLO_3$ chemistry is dominated by strong titration consumption under low temperature and high-$NO_2$ conditions. Therefore, mixing layer capping inversion acts as a barrier in FT-to-ML entrainment. In the $MLO_3$-dominated pattern, MLO3 chemistry is dominated by strong photochemical production under high temperature and high-$CH_2O$ conditions. Therefore, mixing layer



capping inversion acts as a barrier in ML-to-FT detrainment.

(6) From the FTO$_3$-dominated to MLO$_3$-dominated pattern, the O$_3$ precursor CH$_2$O (NO$_2$) substantially increases (decrease) in Beijing, but increases (decreases) slightly in Hong Kong. Over both megacities, the CH$_2$O increment is larger in the upper ML, whereas the NO$_2$ decrement is larger in the lower ML. Such changes in O$_3$ precursors push O$_3$ production sensitivity away from the VOC-limited regime (towards higher

NO$_x$ sensitivity) and facilitate net production of O$_3$ via photochemical reactions, particularly in the upper ML.

Comparing the above results with previous northern mid-latitude observations (Petetin et al., 2018), lower troposphere O$_3$ variability over high-emission Chinese megacities is more likely controlled by O$_3$-related chemical

processes, including titration consumption and photochemical production. From our comparison of Beijing and Hong Kong, lower troposphere O$_3$ variability in China is not only subject to precursor emissions, but also reflects local topographical and meteorological characteristics. Therefore, to achieve comprehensive understanding of lower troposphere O$_3$ variability in China, more ozonesonde observations over more sites are needed in the future.

**Data availability**

Ozonesonde data for Beijing are available from the first author upon reasonable request (lzhiheng118@163.com).
Ozonesonde data for Hong Kong are available at https://woudc.org/home.php?lang=en.
OMI-based ozone precursor data are available at https://disc.gsfc.nasa.gov/.
EAC4-based ozone precursor reanalysis data are available at https://ads.atmosphere.copernicus.eu/.


**Author contributions**

ZL and SF designed the research. ZL organized and wrote the manuscript. MG and JQ edited the manuscript. JS contributed to satellite data analysis and code writing. JZ and YP contributed to ozonesonde observations in Beijing. All authors contributed to the revision of the manuscript.


**Competing interests**

The contact author has declared that none of the authors has any competing interests.

**Disclaimer**

Publisher's note: Copernicus Publications remains neutral with regard to jurisdictional claims in published maps and institutional affiliations.

**Financial support**

This work is supported by the major project of Basic and Applied Basic Research project of Guangdong Province

(grant no. 2020B0301030004), the Key-Area Research and Development Program of Guangdong Province (grant no. 2020B1111360003), the National Natural Science Foundation of China (grant no. 42293321 and 41975181).

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
