# Peer review of "Mixing-layer-height-referenced ozone vertical distribution in the lower troposphere of Chinese megacities: Stratification, classification, meteorological, and photochemical mechanisms"

_EGUsphere, 2023_

## Referee Comment (RC1)

Review of "**Mixing-layer-height-referenced ozone vertical distribution in the lower troposphere of Chinese megacities: Stratification, classification, meteorological, and photochemical mechanisms**" by Liao et al.

In this paper, Liao and the co-authors presented a new way to investigate vertical ozone variation in lower troposphere below ~5 km, which is to scale the ozone vertical profile by the mixing layer height (H). In this way, the authors separate the ozone profiles into two parts: in the mixing layer (ML) and free troposphere (FL), taking ozonesonde observation in Beijing and Hong Kong from 2000-2022. Therefore, the authors are able to obtain some new understanding of ozone vertical variation in the two layers, in the interface of the two layers, and at the surface. Otherwise, such understanding may be lost if the conventional height scale is used. Using the H-referenced scale, the authors further characterized three types of vertical ozone profiles in the lower troposphere: FL-ozone dominated, ML-ozone dominated, and uniform distribution. Through meteorological and photochemical interpretations, the authors attempted to explain the vertical ozone variations in the three types of profiles.

Understanding vertical ozone variation is of importance to ozone pollution management at the surface. This paper provides some new understanding in this regard. The topic is also suitable to this journal. The paper overall read well. I recommend acceptance of this paper and provide the authors with the following suggestions for them to consider when revising their paper.

The authors did a good job in the first part of their paper (Figs. 1-7). The second part is also well-written, but a more in-depth investigation is necessary. For example, the meteorological interpretations for the three types of vertical profiles (Fig. 7) also contain signals of the seasonal variations in the meteorological variables. In addition, it is unclear how the Asian monsoon and associated large scale vertical motions impact the H-referred ozone profiles in different seasons. In the end, the mechanisms for the three kinds of ozone vertical profiles are not clearly articulated.

Minor points:
Fig.8, please explain the numbers at the top of each panel in the caption.

L501, Surface ozone, not lower tropospheric ozone, is autumn-high/summer low in Hong Kong. Ozone in the lower troposphere below 5 km is also high in spring in Hong Kong.

---

## Author Comment (AC1)

Review of "Mixing-layer-height-referenced ozone vertical distribution in the lower troposphere of Chinese megacities: Stratification, classification, meteorological, and photochemical mechanisms" by Liao et al.

In this paper, Liao and the co-authors presented a new way to investigate vertical ozone variation in lower troposphere below ~5 km, which is to scale the ozone vertical profile by the mixing layer height (H). In this way, the authors separate the ozone profiles into two parts: in the mixing layer (ML) and free troposphere (FL), taking ozonesonde observation in Beijing and Hong Kong from 2000-2022. Therefore, the authors are able to obtain some new understanding of ozone vertical variation in the two layers, in the interface of the two layers, and at the surface. Otherwise, such understanding may be lost if the conventional height scale is used. Using the H-referenced scale, the authors further characterized three types of vertical ozone profiles in the lower troposphere: FL-ozone dominated, ML-ozone dominated, and uniform distribution. Through meteorological and photochemical interpretations, the authors attempted to explain the vertical ozone variations in the three types of profiles.

Understanding vertical ozone variation is of importance to ozone pollution management at the surface. This paper provides some new understanding in this regard. The topic is also suitable to this journal. The paper overall read well. I recommend acceptance of this paper and provide the authors with the following suggestions for them to consider when revising their paper.

The authors did a good job in the first part of their paper (Figs. 1-7). The second part is also well-written, but a more in-depth investigation is necessary. For example, the meteorological interpretations for the three types of vertical profiles (Fig. 7) also contain signals of the seasonal variations in the meteorological variables. In addition, it is unclear how the Asian monsoon and associated large scale vertical motions impact the H-referred ozone profiles in different seasons. In the end, the mechanisms for the three kinds of ozone vertical profiles are not clearly articulated.

Reply: Thank you for your positive and constructive comments. We have carefully considered your suggestions and comments, and made corresponding modifications and explanations as follows:

In order to prevent the disturbance from seasonal signals in the meteorological variables, we made a large adjustment in Section 3.3 from previous annual scale to only polluted season scale (summer in Beijing and autumn Hong Kong). We excluded other seasons because those seasons are dominated by single ozone profile pattern ($FTO_3$-dominated pattern). In contrast, summer in Beijing and autumn in Hong Kong correspond to comparable occurrence of different $O_3$ profile patterns. Therefore, the focus on polluted seasons will lead to a more in-depth understanding of ozone pollution in Beijing and Hong Kong.

**Due to the abovementioned adjustment, we re-plotted figures and re-wrote section**

**3.3**. We added a new chart to characterize the lower-tropospheric ozone in polluted season (Fig. 7 in the revised manuscript) and a new chart to characterize the large-scale meteorological conditions associated with different ozone profile patterns (Fig. 8 in the revised manuscript). Meanwhile, we deleted the previous Figure 9 because the available sample numbers of OMI-based ozone precursors are very limited in summer of Beijing and autumn of Hong Kong.

You suggested that the influences from the Asian monsoon and associated large scale vertical motions should be considered in explaining the different ozone profile patterns. However, we found that the vertical motions played a very limited role in shaping the different lower-tropospheric ozone profile patterns. Two evidences are listed as follows: 1) In summer Beijing or autumn Hong Kong, the three ozone profile patterns are characterized by significant difference in MLO3 concentrations but similarity in FTO3 concentrations (Fig. 7 in the revised manuscript). The similar FTO3 concentrations indicate that downward transport of O3-rich air masses from upper level is not a decisive factor for the formation of different ozone profile patterns. 2) Vertical velocity had no significant difference among the different ozone profile patterns (Seeing Figure shown below). The large-scale meteorology (e.g., the Asian monsoon) may play a role through changing photochemistry-related local meteorology, the horizontal transport of ozone and its precursors, rather than through modulating the vertical exchange of $O_3$ between free troposphere and mixing layer. That is to say, it is the mixing layer ozone production (weak or strong production) that shapes the different ozone profile patterns. So, more attentions were paid to photochemistry-related conditions (including local and regional meteorology, ozone precursors and ozone production sensitivity) in the revised manuscript.

[Figure]

Figure S1. Composited vertical velocity at 850 hPa according to different ozone profile patterns in (a) summer of Beijing and (b) autumn of Hong Kong. The red boxes indicate the locations of Beijing and Hong Kong.

Minor points:

Fig.8, please explain the numbers at the top of each panel in the caption.

Reply: Thank you for pointing out our carelessness. The numbers denote the sample numbers. We clarify it in the revised manuscript.

L510, Surface ozone, not lower tropospheric ozone, is autumn-high/summer low in Hong Kong. Ozone in the lower troposphere below 5 km is also high in spring in Hong Kong.

Reply: Thanks for pointing out this false description. We correct it in the revised manuscript.

---

## Author Comment (AC2)

Overview

The submitted manuscript deals with ozone vertical distribution in the lower troposphere over the Chinese megacities of Beijing and Hong Kong. I think that it is an interesting study giving a further insight into the complex meteorological and chemical phenomena associated with the ozone vertical distribution, especially using the Mixing-layer-height-referenced ozone values. I find that the manuscript is generally scientifically sound and well-presented, and, in my opinion, it deserves publication after the recommendations listed below are considered.

Reply: So glad to receive your positive and constructive comments. We have carefully considered your suggestions and comments, and made corresponding modifications and explanations.

General comments

The separation of the profiles based on Mixing-layer-height-referenced ozone into groups that are mainly influenced from the free troposphere (FTO3-dominated) or the boundary layer (MLO3-dominated) over the two examined Chinese megacities is very interesting. Since it is known that ozone pollution might affect large geographical areas through transport and photochemistry in combination to the fact that ozone is regarded as a priority pollutant in China, I think that it would be useful to examine the synoptic meteorological characteristics associated with each group of vertical ozone profiles, especially those influenced from the free troposphere and the boundary layer respectively. So, I would suggest plotting the corresponding composite charts of some meteorological parameters (geopotential height, vector wind, omega, specific humidity) in the boundary layer – free troposphere (850-700 hPa) of the highest ozone profiles (e.g. $10^{th}$ percentile) which are either FTO3-dominated or MLO3-dominated respectively at least for summer, based for example on the categorization of profiles presented on Fig. 5 (e.g. see papers Kalabokas et al, ACP, 2013 and Kalabokas et al., Tellus, 2015, dealing with vertical ozone profiles in the Mediterranean).

Reply: Thanks for your suggestion regarding the examination of synoptic meteorological characteristics associated with different ozone profile patterns. Just as you mentioned, ozone pollution in China is usually a regional problem accompanied by regional-scale transport and photochemistry. In the revised manuscript, we added a description of large-scale meteorology (including geopotential height, vector wind at 850 hPa) associated with different ozone profile patterns (Fig. 8 in the revised manuscript).

You also suggest that attention needs to be paid to those profiles influenced from the free troposphere and the boundary layer respectively. However, we found that the distinct ozone profile patterns are mainly shaped by boundary layer ozone production (weak production or strong production), rather than vertical motions. In summer Beijing or autumn Hong Kong, the three ozone profile patterns are characterized by significant difference in $MLO_3$ concentrations but similarity in $FTO_3$ concentrations (Fig. 7 in the

revised manuscript). The similar FTO$_3$ concentrations among the different patterns indicate that downward transport of O$_3$-rich air masses from upper level is not a decisive factor for the formation of different ozone profile patterns. Meanwhile, we examined the vertical velocity at 850 hPa in different ozone profile patterns. No significant differences are found for the different patterns, indicating a limited role of downward transport. So, we did not further add the composite charts of meteorological parameters in the boundary layer – free troposphere (850-700 hPa) of the highest ozone profiles (e.g. 10th percentile) as presented in references (Kalabokas et al, ACP, 2013 and Kalabokas et al., Tellus, 2015). We believe that the composite charts for each of different ozone profile patterns (Fig. 8 in the revised manuscript) are enough to describe the synoptic controls. In the revised manuscript, more attentions were paid to the examination of photochemistry-related processes, including the favorable local meteorology and regional precursor transport.

[Figure]

Figure S1. Composited vertical velocity at 850 hPa according to different ozone profile patterns in (a) summer of Beijing and (b) autumn of Hong Kong. The red boxes indicate the locations of Beijing and Hong Kong.

Different from previous studies in the Mediterranean, high emissions of ozone precursors in China causes wide-range variability in MLO$_3$ production under the varying synoptic meteorological conditions (favorable or unfavorable). We noted that your mentioned references in the Mediterranean defined the lower troposphere as the height range of 1.5-5.0 km, excluding the atmospheric mixing layer (< 1.5 km). However, the lower troposphere defined in our study included the atmospheric mixing layer. The inclusion of

atmospheric mixing layer makes the wide-range MLO$_3$ variability a decisive factor for the formation of different lower-tropospheric ozone profile patterns in our study. As we mentioned in manuscript, the mixing layer top inversion is so strong to inhibit the ozone exchange between free troposphere and mixing layer. Though that vertical exchange may still works through micro-scale turbulence mixing near the mixing layer top, a complete understanding of this process is not possible given current data and knowledge.

Compared with the lower troposphere focused in this study, our previous studies had discussed the effects of vertical motions on the whole tropospheric ozone profile patterns in Hong and Beijing (Liao et al., 2021; Zeng et al 2023). We found that dynamical processes (e.g., stratospheric intrusion) play a very important role in the upper troposphere. However, the stratospheric ozone intrusion had a very limited influence (or weak signal) on ozone in the atmospheric boundary layer. Recently, Chen et al. (2022) analyzed an event of rapid nocturnal O$_3$ enhancement (NOE) observed on 31 July 2021 at surface level in the North China Plain and proposed transport of substantial stratosphere ozone to the surface by Typhoon In-fa followed by downdraft of shallow convection as the mechanism of the NOE event. However, Zheng et al. (2023) argued that the NOE was not caused by typhoon-induced stratospheric intrusion but originated from fresh photochemical production in the lower troposphere.

Liao, Z., Ling, Z., Gao, M., Sun, J., Zhao, W., Ma, P., et al. Tropospheric ozone variability over Hong Kong based on recent 20 years (2000–2019) ozonesonde observation. Journal of Geophysical Research: Atmospheres, 2021,126, e2020JD033054. https://doi. org/10.1029/2020JD033054.

Zeng, Y., Zhang J., Li, D., Liao, Z., et al. Vertical distribution of tropospheric ozone and its sources of precursors over Beijing: Results from ~ 20 years of ozonesonde measurements based on clustering analysis. Atmospheric Research 284 (2023) 106610.

Chen, Z., Liu, J., Qie, X., Cheng, X., Shen, Y., Yang, M., Jiang, R., and Liu, X.: Transport of substantial stratospheric ozone to the surface by a dying typhoon and shallow convection, Atmos. Chem. Phys., 22, 8221–8240, https://doi.org/10.5194/acp-22-8221-2022, 2022.

Zheng, X., Yang, W., Sun, Y., Geng, C., Liu, Y., and Xu, X.: Comment on "Transport of substantial stratospheric ozone to the surface by a dying typhoon and shallow convection" by Chen et al. (2022), EGUsphere [preprint], https://doi.org/10.5194/egusphere-2023-2336, 2023.

The charts might cover the geographical area between the two megacities, which is densely populated and with important air pollutant emissions. This would give a good idea of the geographical extend of the large-scale tropospheric ozone subsidence observed during summer months over Beijing or respectively to the uplifting of boundary layer air towards the free troposphere observed over Hong Kong.

Reply: Thank you for this constructive suggestion. We made the charts covering the Eastern China (Figs. 8 in the revised manuscript). However, these charts are more used to reflect large-scale transport of ozone and its precursors, rather than ozone subsidence.

I think that the above supplementary work would strengthen the results of the paper by giving a further insight of the local or regional nature of the meteorological phenomena associated with FTO3-dominated or MLO3-dominated ozone profiles observed over Beijing and Hong Kong. In addition to their scientific importance, these results might be useful for the development of more elaborated ozone pollution abatement strategies in the examined areas of China.

Reply:Thank you for your constructive suggestions again. The corresponding revisions according to your suggestions have substantially strengthened the results by giving a further insight of the local or regional nature of the meteorological phenomena associated with FTO3-dominated or MLO3-dominated ozone profiles observed over Beijing and Hong Kong. We believe that the results will be useful for the development of more elaborated ozone pollution abatement strategies in the whole eastern China.